# The Functional Biogeography of eDNA Metacommunities in the Post-Fire Landscape of the Angeles National Forest

**DOI:** 10.3390/microorganisms10061218

**Published:** 2022-06-14

**Authors:** Savanah Senn, Sharmodeep Bhattacharyya, Gerald Presley, Anne E. Taylor, Bruce Nash, Ray A. Enke, Karen B. Barnard-Kubow, Jillian Ford, Brandon Jasinski, Yekaterina Badalova

**Affiliations:** 1Department of Agriculture Sciences, Los Angeles Pierce College, 6201 Winnetka Avenue, PMB 553, Woodland Hills, CA 91304, USA; fordjm7164@student.laccd.edu (J.F.); jasinsbk9887@student.laccd.edu (B.J.); badaloya@lavc.edu (Y.B.); 2Environmental Sciences Graduate Program, Oregon State University, Corvallis, OR 97331, USA; sharmodeep.bhattacharyya@oregonstate.edu (S.B.); gerald.presley@oregonstate.edu (G.P.); anne.taylor@oregonstate.edu (A.E.T.); 3Department of Statistics, Oregon State University, Corvallis, OR 97331, USA; 4Department of Wood Science & Engineering, Oregon State University, Corvallis, OR 97331, USA; 5Department of Crop and Soil Sciences, Oregon State University, Corvallis, OR 97331, USA; 6DNA Learning Center, Cold Spring Harbor Laboratory, Cold Spring Harbor, NY 11724, USA; nash@cshl.edu; 7Department of Biology, Center for Genome & Metagenome Studies, James Madison University, Harrisonburg, VA 22807, USA; enkera@jmu.edu (R.A.E.); barnarkb@jmu.edu (K.B.B.-K.)

**Keywords:** DNA sequencing, functional diversity, molecular ecology

## Abstract

Wildfires have continued to increase in frequency and severity in Southern California due in part to climate change. To gain a further understanding of microbial soil communities’ response to fire and functions that may enhance post-wildfire resilience, soil fungal and bacterial microbiomes were studied from different wildfire areas in the Gold Creek Preserve within the Angeles National Forest using 16S, FITS, 18S, 12S, PITS, and COI amplicon sequencing. Sequencing datasets from December 2020 and June 2021 samplings were analyzed using QIIME2, ranacapa, stats, vcd, EZBioCloud, and mixomics. Significant differences were found among bacterial and fungal taxa associated with different fire areas in the Gold Creek Preserve. There was evidence of seasonal shifts in the alpha diversity of the bacterial communities. In the sparse partial least squares analysis, there were strong associations (r > 0.8) between longitude, elevation, and a defined cluster of Amplicon Sequence Variants (ASVs). The Chi-square test revealed differences in fungi–bacteria (F:B) proportions between different trails (*p* = 2 × 10^−16^). sPLS results focused on a cluster of Green Trail samples with high elevation and longitude. Analysis revealed the cluster included the post-fire pioneer fungi *Pyronema* and *Tremella*. *Chlorellales* algae and possibly pathogenic *Fusarium* sequences were elevated. Bacterivorous *Corallococcus*, which secretes antimicrobials, and bacterivorous flagellate *Spumella* were associated with the cluster. There was functional redundancy in clusters that were differently composed but shared similar ecological functions. These results implied a set of traits for post-fire resiliency. These included photo-autotrophy, mineralization of pyrolyzed organic matter and aromatic/oily compounds, potential pathogenicity and parasitism, antimicrobials, and N-metabolism.

## 1. Introduction

Wildfires have continued to increase in frequency and severity in Southern California due in part to climate change; furthermore, the size and intensity of fires has increased since 1950 [1]. To gain a further understanding of microbial soil communities’ response to fire and functions that may enhance post-wildfire resilience, soil fungal and bacterial microbiomes were studied from different wildfire areas in the Gold Creek Preserve within the Angeles National Forest using 16S, FITS, 18S, 12S, PITS, and COI (Cytochrome Oxidase I) amplicon sequencing.

Dispersal is a factor in the evolution of metacommunities [2]; in the setting of the study site, Canyon relief, creek flow, and wind erosion in the Gold Creek Preserve aid in dispersal of microbes. Wind erodibility is a part of soil classification [3]. Habitat isolation, distance from the forest edge, or remoteness is also a factor that shapes community structure in the biogeographical context [2].

Metacommunity ecology has focused on species attributes, including eukaryotic host distributions, and landscape ecology focuses on site attributes [2]. This study attempts to integrate both sets of attributes into a joined model for the assemblages of organisms that are present in high relative abundance at various spatial coordinates. Antibiotic resistance represents a rapid evolutionary process which dons novel ecological functions to bacteria and modifies the competitive hierarchy [2]. Coexistence networks or clustered image maps [4,5] may shed light on the evolution of groups of microorganisms in the context of the study location and extent.

Heat from an intense wildfire would be expected to sterilize soil [6,7] and favor taxa that metabolize pyrolyzed organic matter, such as *Pyronema* fungi [6]. Wetter soil would be expected to increase the impact of heat on soil, along with fire intensity and duration [7]. Microbial biomass is characteristically lowered following a wildfire [8,9], and recovery of soil communities may span decades [9]. Microbial diversity would be expected to decline with the number of fires, as noted by Bowd et al. [10]. However, soil erodibility is enhanced post-fire [11], which would aid in the dispersal of organisms. Additionally, low intensity fires may not significantly decrease mycorrhizal populations, fire effects are expected to be less in xeric environments, and microbes in the substratum are more resistant to burns [7]. Oliver et al. [8] reported no shifts in soil physical properties following low-intensity burns, although fungal taxa did shift in composition.

Fungi can be readily identified in soil samples with fungi internal transcribed spacers (FITS) markers. Vegetation communities have been strong predictors of bacterial and fungal community responses to fires in boreal forests in Canada [9]. Plant environmental DNA (eDNA) in the soil can be identified with PITS (plant internal transcribed spacers) markers, although visual inspection of the site may be similarly informative. Typically, bacteria are identified with the 16S marker.

Although bacteria and fungi were of primary interest, other microorganisms such as algae and protists were also detectable with this assay, as well as nematodes and arthropods. Protists have received little attention in studies of microbial communities [12]; this emphasizes the importance of including markers such as 18S and COI for identification of eukaryotes in metabarcoding datasets. Predators control the size of populations, are interdependent with microbes, play a role in designing the community structure, and influence spatial heterogeneity through grazing [2]. These eukaryotic organisms may also have symbiotic associations that help to shape the persistence of symbionts even when the association no longer benefits both parties [2]. Furthermore, rapid establishment of photosynthetic organisms is common after fire, and a decrease in the F:B ratio is common after fire [7].

This study will contribute to the resolution of the important ecological inquiry questioning the limits to which microbial metacommunity diversity and functions are resistant to short and long-term perturbations, how much functional redundancy is present in microbial communities, and how major shifts in environmental conditions may affect co-occurring groups of microorganisms [13].

We hypothesized that the Blue Trail, which is a wildlife corridor and was only burned in one recent fire, would have the highest alpha diversity of fungi and the highest fungi–bacteria (F:B) ratio. We hypothesized that trails would have a differential abundance of taxa associated with their fire histories. We hypothesized that the Blue Trail would have the highest alpha diversity of fungal taxa when contrasted with the other trails, which had different fire histories. The Red Trail was burned in the 2009 Station Fire combined with the Creek Fire, and the Creek Fire and 2016 Sand Fire affected the Green Trail.

## 2. Materials and Methods

Data was generated from two rounds of soil sampling. Gold Creek Round 1 (20 samples) were taken in December 2020, and Round 2 (18 samples) were taken in June 2021. Data available from Round 1 includes 16S, 18S, 12S, FITS, COI and PITS. The paper will focus on soil microbes. December 2020 and June 2021 data were analyzed using DNA Subway Purple Line [14], ranacapa [15], stats, vcd, and mixomics [5,16]. Ranacapa was used for multivariate ANOVA and alpha diversity. R stats was used for Chi-square test of proportions. Mixomics was used to conduct sparse partial least squares analysis (sPLS). Samples were gathered from the Blue Trail wildlife corridor which was burned in the 2017 Creek fire. The Red Trail burned in the 2009 Station Fire combined with the Creek Fire, and the Creek Fire and 2016 Sand Fire affected the Green Trail. The information for each of the fires is given in Table 1.

There are two sets of soil conditions present between the sites, based on USDA historical data [3]. Both of the soil classifications in the sampling area had low organic matter (OM) from 0 to 10 cm. The Trigo-Modesto-San Andreas classification encompassed the southern half of the Preserve, including the Blue Trail and the Red Trail. The Caperton-Trigo classification was mainly located the Green Trail and is represented by Map unit 54. The sites differ in their range of slope, pH, cation exchange capacity (CEC), texture, and capacity for capillary water, as shown in Table 2. Since the Caperton-Trigo classification was coarser, it had less capacity for available water, and was well-drained. The Caperton-Trigo area also had a slightly acidic pH based on historical data, and a higher cation exchange capacity, which one would expect to be favorable to plants despite the diminished water availability. Furthermore, the samples associated with the Caperton-Trigo complex corresponded to a lower soil wind erodibility index (48 T/ac/year) regardless of its higher maximum slope percentage, when compared with the Trigo-Modesto-San Andreas complex (56 T/ac/year) [3].

Soil samples were taken from plant rootzones of *Quercus agrifolia*, *Eriodictyon crassifolium*, *Dendromecon rigida*, and *Arctostaphylos glauca*. The *Q. agrifolia* had burned in different combinations of fires, and the thickleaf Yerba Santa and Tree Poppy, which require scarification to germinate, established large stands at the sampling sites after the 2017 Creek Fire and 2016 Sand Fire, respectively. The manzanita trees resprouted from trees that had been established after the Station Fire and regenerated following the 2017 Creek Fire. These plants were of interest since they produce secondary metabolites, such as tannins, coumarins, protopine, and essential oils. Coumarins and terpenes influence the composition of microbial communities in the rootzone [18]; microbes regulate plant secondary metabolite production by enhancing gene expression or through horizontal gene transfer [19].

For Round 1, three samples were retrieved within a 1-foot radius of one another, and the DNA was pooled prior to library construction for each marker. Three samples were taken from three individuals of the same plant species in proximity of each marker for Round 2. The soil sampling maps for winter and summer are displayed in Figure 1. Sample metadata for summer 2021 and winter 2020 has been provided in Table 3 and Table 4, respectively. There were two rounds of sampling, during winter and during summer (38 samples total) with 16S, FITS, 18S, 12S, PITS, and COI amplicons. The samples were collected in sterile cryotubes. For Round 1 samples, the DNA was extracted at University of California, Los Angeles and sequenced by University of California, Santa Cruz Genomics Institute [20]. For Round 1, six molecular markers were amplified and sequenced with Illumina barcode adapters at 35,000 paired reads each. Quality control was performed in QIIME2; Cutadapt was used to remove Illumina adaptor sequences; DADA2 was used for quality score trimming and identification of unique ASVs. Taxonomies were assigned to Amplicon Sequence Variants with an 95% likelihood cutoff from the CRUX database. A GreenGenes classifier was used. Each marker dataset was outputted into an ASV (Amplicon Sequence Variant) table for downstream analysis using the Anacapa toolkit [21].

Round 2 Data consists of 16S amplicons only. For Round 2, DNA was extracted with the Qiagen (Hilden, Germany) Power Soil DNA kit and sent to James Madison University for 16S amplification, library preparation [22,23], and pooled 16S amplicon NGS on the Illumina (San Diego, CA, USA) MiniSeq platform. The V4 region of the bacterial 16s rRNA gene was amplified and barcoded for each sample using the primers developed by Kozich et al. (2013). Samples were screened for successful amplification on an agarose gel and pooled. A double-sided bead cleanup was carried out to remove primer-dimers and a low amount of off-target larger PCR products. Quality and concentration of the pooled library was checked using a Bioanalyzer (Agilent, Santa Clara, CA, USA) and NEB’s Library Quant Kit for Illumina. The library was then sequenced on a MiniSeq using a mid-output reagent cartridge. Before loading, the library was combined with Ilumina’s PhiX control (30:70 16s:PhiX) to ensure a high-quality run despite the low diversity of the 16s library. A dual indexing strategy was used with these primers which used Kozich et al.’s approach and the Schloss primers: Forward: GTGCCAGCMGCCGCGGTAA. Reverse: GGACTACHVGGGTWTCTAAT [24,25]. DNA Subway was used; the Purple Line analysis implemented DADA2 [26] and QIIME2 [27,28] for quality control, alpha rarefaction, and to output the ASV table for taxonomic diversity analyses.

The methods used to identify the most important features employ a similar strategy as Dwiyanto et al. [29], which included PERMANOVA, sparse partial least squares, differential abundance analysis; we also considered the functional implications using predicted functions and references to known microbial functions. There were less robust functional databases developed for fungi than bacteria, and protists are primarily characterized by morphology [12] so there was less information available about their function in the literature. Formal hypothesis testing was carried out in DESeq2 and visualized with SystempipeR. The Round 1 and Round 2 16S Fastqs were compared using EZBioCloud Metagenomic Taxonomic Profiling applications.

To examine other factors besides the fire history which may have been associated with the most abundant microbes in the study, partial least squares clustering was performed. sPLS has been used for a range of applications which have included genomic selection in cattle breeding [30], data integration for expression studies and eQTL (expression quantitative trait loci) mapping [31,32], has helped establish relationships between soil fungal diversity and the concentration of chlorinated pollutants [33], and has uncovered differential abundance of microbial WGS shotgun metagenomic sequences [34].

Partial least squares has been considered a quasi-supervised approach. It has been particularly useful for genomics and environmental sampling because it is a solution to the *p* > n problem. That is to say, there are relatively few samples compared to the number of features in the DNA sequencing dataset. Essentially, successive regressions are carried out via projection onto latent constructs to unveil hidden biological effects [35].

Next, variable selection was carried out to reveal the most important features in a large dataset which can be obscured by an overabundance of features. The optimal number of features is tuned for each of the selected number of components using k-fold cross-validation with the purpose of minimizing the root mean square error of estimation [5,30]. In genomic selection efforts, the predictive error is minimized [30]. The model was further filtered using regression coefficients [5,16,35].

The dataset was merged to reduce the number of necessary steps and was intended to give a more wholistic picture of the microbiome, including which bacteria, archaea, plants, fungi, and protists are associated with one another and their potential interactions. The Round 1 data from 16S, FITS, PITS, 18S, and COI were concatenated, and zero-sum columns were removed with R janitor [36]. The package mixomics was used to conduct sparse partial least squares analysis, which is useful when there are a number of multicollinear features, as in this case.

## 3. Results

The main results showed significant differences in bacterial and fungal taxa associated with different fire areas in the Gold Creek Preserve. There were changes in fungal taxa associated with soil samples that were affected by the Creek Fire in combination with the 2009 Station Fire, 2016 Sand Fire, or the 2017 Creek Fire alone. The computation of the Abundance-based Coverage Estimator (ACE) index accounted for the number of rare and abundant OTUs [37]. The highest fungal to bacterial ratio was related to the area that had been affected by the two most recent fires, which was the Green Trail. However, there was in fact a trend toward the highest alpha diversity of fungi residing in the Blue Trail wildlife corridor.

Plant species was not a significant factor in determining the alpha or beta diversity of the communities. However, in the sparse partial least squares analysis, there was evidence of moderate associations (r > 0.5) between the type of plant, topography, soil vs. sediment, soil classification, latitude, and a suite of prokaryotic and eukaryotic taxa. There were strong associations (r > 0.8) between longitude, elevation, and a defined cluster of Amplicon Sequence Variants (ASVs).

The results of the permutational ANOVA for the summer bacteria sequences indicated that the Trail variable was associated with differences in beta-diversity between samples (*p* = 0.04). The largest differences were between the Red Trail and the Blue Trail in the winter samples. In the December samples, there was also a marginally significant result for higher alpha diversity in bacterial communities associated with the Blue Trail when contrasted with the Red Trail, based on a post hoc Tukey Test (*p*-adjusted = 0.09). The Red Trail Samples were associated with a significantly lower number of observed bacteria species when contrasted with the Green Trail. This evidence suggested that the higher fire intensity of the Station Fire combined with the recent burn of the Creek Fire influenced the abundance of fungal sequences amplified by the FITS marker. The Blue Trail was associated with significant differences in 16S beta diversity when contrasted with the Red Trail, based on the analysis in Ranacapa.

The Blue Trail samples taken as a whole, which had burned in the Creek Fire, were associated with the most observed species for fungi. The salty area at the head of the canyon on the Green Trail exhibited the least species richness for fungi. Overall, however, the Red Trail had the lowest number of observed species, in general, as Figure 2 has revealed. In the summer samples, the Trail variable was a significant factor associated with species richness (*p* = 0.029). There were significant differences in the observed number of species between the Red Trail and the Green Trail (*p*-adjusted = 0.022). This was a shift from the winter samples, where differences were detected between the species richness of bacteria between the Red Trail and the Blue Trail. There were significant differences in the summer samples in terms of beta diversity between the Blue Trail and the Green Trail as well (*p*-adjusted = 0.04), based on PERMANOVA.

Plant rootzone was not significantly associated with alpha/beta diversity of the communities based on PERMANOVA (permutational multivariate analysis of variance) and observed species. However, in sPLS, moderate associations (r > 0.5) showed between plant, topography, soil/sediment, soil classification, latitude, and suites of taxa. Strong associations (r > 0.8) existed between longitude, elevation, and clusters of taxa. The sPLS was useful for selecting relevant features. sPLS visualizations revealed evidence of overdispersion, shown in Figure 3.

In the December samples, the ratio of bacterial to fungal sequences was calculated. A Chi-square test of proportions was carried out. The null hypothesis was that there was no difference in the proportions of bacteria to fungi between the three Trails. The F:B ratio for the Blue Trail was 2.6, which was between the range for subtropical forest and desert, based on published datasets [38]. The F:B ratio for the Red Trail was 3.18, which was similar to cropland [38]; the ratio for the Green Trail was 3.47, which was similar to shrubland and grassland [38]. The results showed that all three groups differed from one another in terms of their bacteria to fungi ratios (*p* = 2.2 × 10^−16^). These results suggested that the effects of repeated burns favor fungi. Interestingly, based on the 18S data from December, sediment samples associated with a higher relative abundance of arthropods, except for the low creek sample. The *Rhagidiidae* predatory mite was an important taxon according to the sPLS results, as shown in Figure 4. Ascomycota sequences had the highest relative abundance, followed by Basidiomycota. The Mucoromycota sequences were mainly present in the Blue Trail soil samples. In Whitman et al., Mucoromycota increased at sites with higher burn severity [9].

The 16S data also showed a differential abundance of bacteria during the second round of sampling on the Green Trail. Higher alpha diversity was observed in the second round of 16S samples taken during the warm season in June, versus in the first round of samples from the winter. There was no significant difference in beta diversity. It was also be considered that the two sets of samples were processed and sequenced on separated runs, which accounts for some of the variation. Nevertheless, the inspection of the negative controls in both datasets suggested that were was little contamination in the DNA isolates or amplicons.

sPLS was sensitive to outliers; results focused on a cluster of Green Trail samples with high elevation and longitude, shown in the top cluster of Figure 4. Analysis revealed the cluster included the post-fire pioneer fungus *Pyronema* [6,39], *Tremella,* and *Strobiloscypha,* which produce strobiloscyphone antimicrobials [40]. *Chlorellales* algae, toxic *Leptiota,* and potentially pathogenic *Fusarium* sequences were elevated. Bacterivorous *Corallococcus,* which secretes antimicrobials [41], and bacterivorous flagellate *Spumella* [42] were associated with the cluster. Plant-associated *Rhogostoma,* of the *Cercozoa* amoebae [43], had a moderately positive association with elevation and longitude and was associated with the soil organism grouping rather than the sediment grouping. This makes sense, since they are extremely abundant in terrestrial ecosystems [43]. 

As evidenced in Figure 4, each cluster also had possible plant pathogens, such as *Phoma, Pyrenochaetopsis* and *Microbotrymycetes,* saprophytes such as *Penicillium* and *Tetracladium*, mycorrhizal fungi such as *Genabea* and *Sporidesmium,* and ammonia oxidizers such as *Nitrosarcheum* and *Planctomyces*. Coprophilous fungi, which colonize dung, were also represented. Parasites such as *Acanthamoeba* were also characterized in each cluster. *Mortierella*, of the Mucoromycota, which are noted for their production of polyunsaturated fatty acids, were associated with a cluster of taxa that were negatively associated with elevation and latitude. Mucoromycota were also shown to be differentially abundant on the Blue Trail, as noted in Table 5 and Table 6, along with other lipid accumulators *Umbelopsidaceae* and *Umbelopsis* sp. Polyketide and antibiotic producers such as *Minuisphaera* and *Neosetophoma* were also broadly distributed, which matches up with the results in Table 5 and Table 6 from the DESeq2 [44] analysis, where different *Cladorrhinum* sp. sequence variants were differentially abundant on each trail. Additionally, shown in Figure 4, each cluster of taxa had autotrophic organisms characterized, such as red, green, golden, or brown algae, or protists which are symbiotic with *Cyanobacteria*. *Gemmatimonadetes* autotrophic bacteria were differentially abundant on the Green Trail and Blue Trail, as noted in Table 7 and Table 8. The Red Trail was associated with methanogens and methanotrophs.

There were no highly significant differences in functional diversity between the different trails in the inferred functional analysis in EZBioCloud MTP [52] based on adjusted *p*-values. The suggested that there is functional redundancy between the separate soil communities, which allowed the soil ecosystem functions to be carried out, but within different soil and fire history environments. There were some interesting trends that are worth pointing out which had significant *p*-values < 0.05 but did not pass multiple testing with FDR = 0.05.

As shown in Figure 5, Assimilatory nitrate reductase predicted functions were elevated the Green Trail for the Round 1 samples; multidrug/chloramphenicol efflux transport protein predicted functions were also elevated on the Green Trail. Elsewhere on the Blue Trail, the Benzene toluene chlorobenzene dioxygenase ferredoxin component was predicted to be elevated in the metagenome, along with Lichenysin synthetase. Beta-lactamase antibiotic resistance functions were elevated for the Blue Trail, as well as Cu and Ag efflux system and multidrug efflux system outer membrane protein, according to the Kruskal–Wallis functional biomarker test in EZBioCloud. Halo acetate dehalogenase was elevated on the Red Trail, according to the predicted functions; Propanediol dehydratase and Stigmatellin polyketide synthase were also expected to be elevated on the Red Trail in the Round 1 samples. Putrescine ornithine dehydratase was elevated on the Green Trail, based on the predicted functional profile in EZBioCloud.

There were similar trends in the Round 2 samples, such as predictions for elevated CAG pathogenicity island protein 4 sequences on the Green Trail vs. Red Trail; there were also predicted elevated functions of coronafacic acid on the Green Trail when contrasted with the Red Trail. Coronafacic acid mimics jasmonic acid in plants and promotes rhizomes. Furthermore, there were elevated functions of coumaroyl quinate monooxygenase on the Green Trail when contrasted with the Red Trail. Monooxygenases are used by microbes to degrade polyaromatic compounds [6]. *Pyronema* fungi which occurred in high abundance also share this function [6].

Taxa showed significant differences at different fire areas in the Preserve. The highest fungi–bacteria (F:B) ratio was related to the Green Trail, burned in 2016 and 2017. The Chi-square test revealed differences in F:B proportions between different trails (*p* = 2 × 10^−16^). There was evidence of overdispersion in the alpha diversity results from both the Round 1 and Round 2 datasets, which indicated that a negative binomial model may be appropriate for differential abundance analysis; the evidence is displayed in Figure 6. It was evident in the partial least squares results as well, where the effect of outliers has a strong influence on the model, and the results focus on the differences between the Green Trail and other trails. There were no *Phytophthora* sequences found in the soil samples except with the COI marker. It was not detected with the FITS or 18S marker. There were several other possible plant pathogens detected including *Alternaria*, which is also a common saprobe.

EZBioCloud Metagenomic Taxonomic Profiles for bacteria associated with the winter versus the summer soil samples is shown in Figure 7. The taxonomies at the phylum level are similar, which was consistent with the beta diversity results for these two groups. The differences between the groups could be accounted for by Cyanobacteria in the winter which were absent in summer. On the other hand, *Gemmatimonadetes* were present in summer and absent in winter, as shown in Figure 7. These changes appeared to represent a seasonal shift in microbial communities. Furthermore, there was a higher relative abundance of *Planctomycetes* in winter, a higher relative abundance of *Verrucomicrobia* in summer, and a higher relative abundance of *Chloroflexi* which are frequently found in hot springs and hypersaline environments [53], in the winter samples.

Further investigation of the taxonomic composition at the class level revealed that multiple classes of bacteria were detected in winter but not in summer, visualized in Figure 8. These classes were *Nitriliruptoria*, *Flavobacteria*, and *Clostridia*. Meanwhile, during summer there were other classes detected that were absent in the winter samples. These classes were *Rubrobacteria, Solibacteres, Oligoflexia,* and *Acidimicrobia*.

## 4. Discussion

The perilous life of plants in fire disturbed areas of the semi-arid forest was observed to be threatened by many factors. Beyond abiotic factors such as fire damage to vascular tissues and increasing frequency and duration of heat and drought, there was evidence that woody plants were also confronted with many possible plant pathogens and parasites. Some notable potential pathogens from the results were *Pythium, Fusarium, Botryosphaeriaceae*, *Microbotrytomycetes*, *Xanthamonadales,* and *Hyphomonas. Pythium* and *Fusarium* are common fungi in bare-root forest nurseries [67]. What was remarkable about the presence of these possible pathogens in the Gold Creek results was the way that the putative pathogens may be controlled by antibiotic-producers that were found in high abundance on the various Trails. There was strong evidence supporting the presence of antibiotic producers such as *Cladorrhinum* sp. in significant numbers, according to our results.

After fire, cyclic organic compounds have been shown to be released from plant material and soil and hydrophobic films cover soil particles [7]. The ability to tolerate and utilize these compounds is a means to survival and resilience in fire disrupted areas. Different groups of organisms with resiliency traits at different combinations of longitude and elevation in the area studied were revealed.

Based on the DNA sequencing results, it appeared that the support system of the recovering forest involved predatory mites, bacterivores, nitrogen fixing cyanobacteria and protist symbionts, algae, ammonia oxidizing bacteria and archaea, *Actinobacteria* which reduce nitrogen, mycorrhizae from Ascomycota, Basidiomycota, and Mucoromycota, and the post-fire pioneers *Pyronema* and other *Pezizales.* In Whitman et al., Mucoromycota increased at sites with higher burn severity [9]. *Pezizales* are known to have increased abundance of sporocarps after a burn [68,69], which would make them poised to colonize when the fire subsides.

Bacterivores such as amoebae were important not only as predators but also due to their role in spreading bacteria. Bacteria which feed on other bacteria were notable for their role in mineralizing the nutrients bound to the body of soil bacteria. Where sugars are scarce, some bacteria and fungi are able to use other energy sources such as amino acids or Nitrogen, while others make their own energy through photosynthesis or facultative photo-autotrophy. Naturally, there were also many soil bacteria in this context that acquire and exhibit antimicrobial drug resistance traits. A few examples of this were the beta-lactamase resistance genes, efflux systems and multidrug resistance.

The high F:B ration on the Green Trail may have been due to the resistant nature of fungal sclerotia [7] which may have had a persistent advantage over bacteria in this setting. Bare soil has also been shown to have a high F:B ratio. Another possible explanation was the winter season of the Creek Fire, combined with the soil classification on the Blue and Red Trail, which would both contribute to the wetness of the soil when the fire took place. Wetter conditions have been shown to contribute to soil sterilization during fires [7]. There would be expected to be a short-lived increase in mobile nitrogen, phosphorus, potassium, calcium and magnesium after a burn [7,11], which could in fact lead to an accumulation of microbial biomass [7]. A flush of CO_2_ immediately during and after a fire, and an increase in metabolic activity in the microbiome post-fire would be expected to occur, after wetting and rewetting of soil in Mediterranean forests according Munoz-Rojas et al.’s Australian study [70]. Furthermore, there were asymmetric changes in the carbon and nitrogen cycles [70]; Rodriguez et al. found similar results in *Quercus ilex* [71]. Elevated CO_2_ has been shown to modify community composition in the wheat rootzone [72]. However, over time leaching, wind, and water erosion leads to a loss of nutrients post-fire [11], and CO_2_ will disperse.

An unexpected result is that the plant species was not a significant factor in determining the alpha or beta diversity of the communities. This is in contrast to what Mataix-Solera et al. suggested [7]; it was suggested that shifts in plant communities would be the drivers of microbial diversity rather than the effects of fire itself. In Whitman et al. [9], there was a trifecta of plant and microbial fire response strategies including fast growers, heat survivors, and post-fire affinity; burn severity was the most important factor. The dispersal mechanisms noted earlier may explain the small number of differentially abundant taxa that were identified in the DESeq2 analysis.

The moderate associations (r > 0.5) between plant, topography, soil/sediment, soil classification, latitude, and suites of taxa partly agreed with previous research [7] and the results of Whitman et al. [9]. Furthermore, we identified different microbial post-fire responders than Whitman et al.’s Canadian study [9], with the exception of *Penicillium*. The plant communities help to configure the clusters of taxa in multi-dimensional space. However, in our study, strong associations (r > 0.8) existed between longitude, elevation, and clusters of taxa, in agreement with Wang et al.’s study on lichen-associated fungi [73].

In Abaya et al.’s study [47], *Cladorrhinum* was demonstrated to be effective against plant pathogens; *Pyrenochaeta* was one of the pathogens controlled. It is interesting because in our results, *Pyrenochaetopsis* was an abundant and ubiquitous pathogen, and *Cladorrhinum* was a differentially abundant fungus associated with the Red Trail and the Green Trail.

As a result of wildfires, aromatic and oily Plant and soil compounds released from fire. Microbial Cyclopropane, benzene, toluene and chloroalkane processing functions are favored. Nitrogen and Phosphorous are liberated from ash. Saprophytic fungi break down organic matter, e.g., lignocellulose. Mushrooms accumulate lipids, gamma linoleic acid, and triacylglyerides. Bacteria produce polyketides such as alkaloids, antimicrobial and anti-insect compounds. Antibiotic resistance and tolerance as a response to selective pressure (Beta lactamase and multidrug resistance). Fungi produce toxins for defense such as amatoxin. Plant pathogen populations are controlled. Plant Pathogens respond to selective pressure by adapting with antibiotic resistance. Nitrogen cycling is carried out by bacteria, archaea, cyanobacteria and their symbionts, and fungi. Nitrogen fixation is key, along with aerobic and anaerobic ammonia oxidizers, saprophytes, and coprophilous fungi which represent the interaction with animal waste and cadavers. These concepts were mapped in Figure 9.

Algae and *Cyanobacteria* have been known as common post-fire pioneers and may quickly colonize degraded ecosystems [7]. However, nitrate and nitrous oxide, oxidized products of nitrogen fixation, are potential sources of groundwater pollution and nitrification-related features should be monitored [74]. In our results, there were highly abundant *Cyanobacteria* and ammonia-oxidizing bacteria and archaea. In Levy-Booth 2014, the author expressed concern about dissimilative nitrogen metabolism [74]. In dissimilative nitrogen metabolism, the nitrate product is excreted into the environment because more is produced than needed to satisfy organismal needs [75].

The Red Trail Samples were associated with a significantly lower number of observed bacteria species when contrasted with the Green Trail. This evidence suggested that the higher fire intensity of the Station Fire combined with the recent burn of the Creek Fire influenced the abundance of fungal sequences amplified by the FITS marker. As Oliver et al. noted [8], the most important factors in soil microbial response to fire are the burn frequency and intensity.

There was evidence of seasonal shifts in the alpha diversity of the bacterial communities, based on the (ACE) index. That was because there were more OTUs detected in the Summer 16S results than the Winter 16S results. Ma et al. found similar results of seasonal community composition shift in the rootzone of hazelnut [4].

## 5. Conclusions

There are a high number of significantly abundant microbes that remain unclassified. This represents a major opportunity for discovery. However, caution is suggested due to the large number of possible plant and animal pathogen and parasite sequences that were discovered to be present in large numbers. Further study of the communities of organisms at this site could lead to the discovery of antibiotics and bioinsecticides, derived from the same organisms that are helping plants in the Angeles National Forest to resist infection.

There was functional redundancy between fungi and bacteria, and between different clusters of samples, that emphasizes the most important functions identified in this study and how they are important to recovery from fire. Clusters of taxa were differently composed but shared similar ecological functions. These results implied a set of traits for post-fire resiliency. These included photo-autotrophy, mineralization of pyrolyzed organic matter and aromatic/oily compounds, potential pathogenicity and parasitism, antimicrobials, digestion of microbes, and N-metabolism.

## Figures and Tables

**Figure 1 microorganisms-10-01218-f001:**
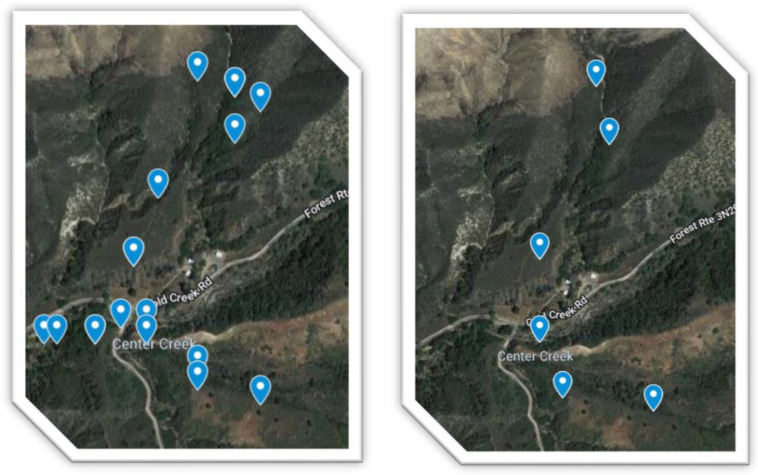
Left to Right: Round 1 soil sampling map (December 2020) and Round 2 soil sampling map (June 2021).

**Figure 2 microorganisms-10-01218-f002:**
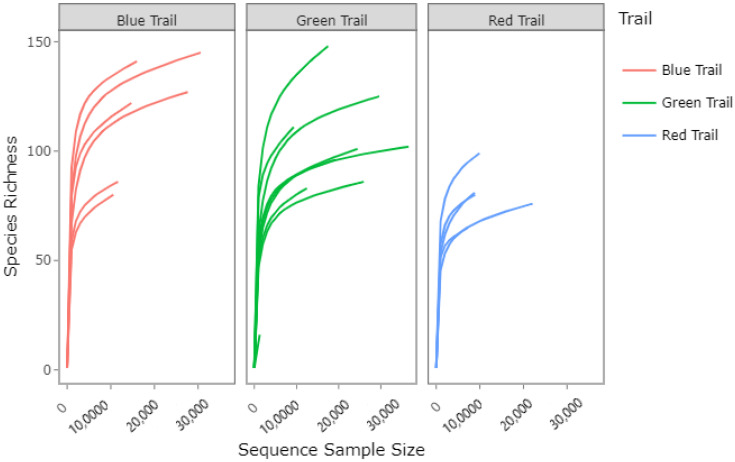
Fungi species richness curves for the three Trail areas of the Gold Creek Preserve.

**Figure 3 microorganisms-10-01218-f003:**
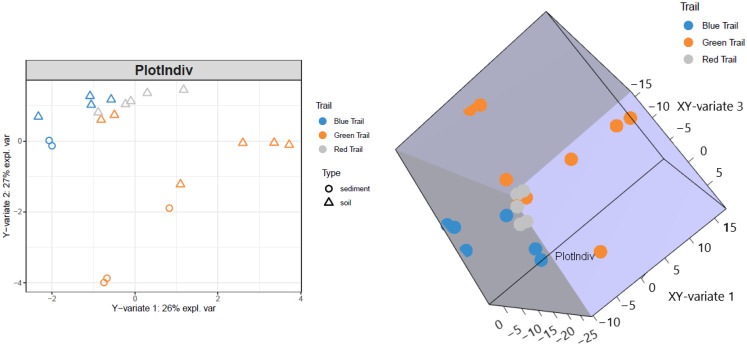
Sparse Partial Least Squares PCA plot of the samples in Y-variate space after optimization of the number of components using the root mean square error minimization. A 3D plot of the final sparse partial least squares PCA depicted in XY-variate space.

**Figure 4 microorganisms-10-01218-f004:**
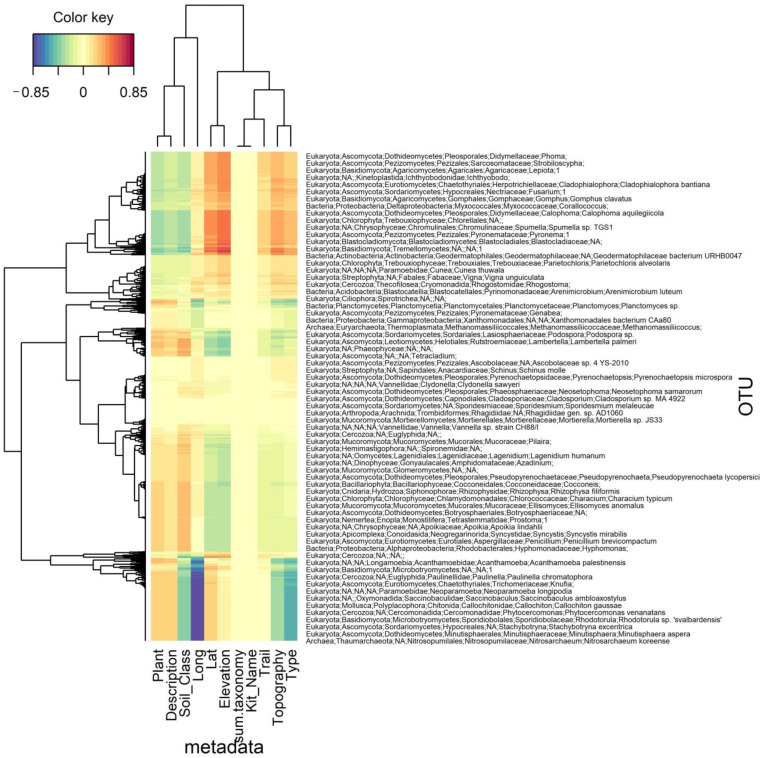
Clustered Image Map. There was a suite of functional traits related to wildfire resiliency that was found to be consistent across clusters of samples in the sPLS results.

**Figure 5 microorganisms-10-01218-f005:**
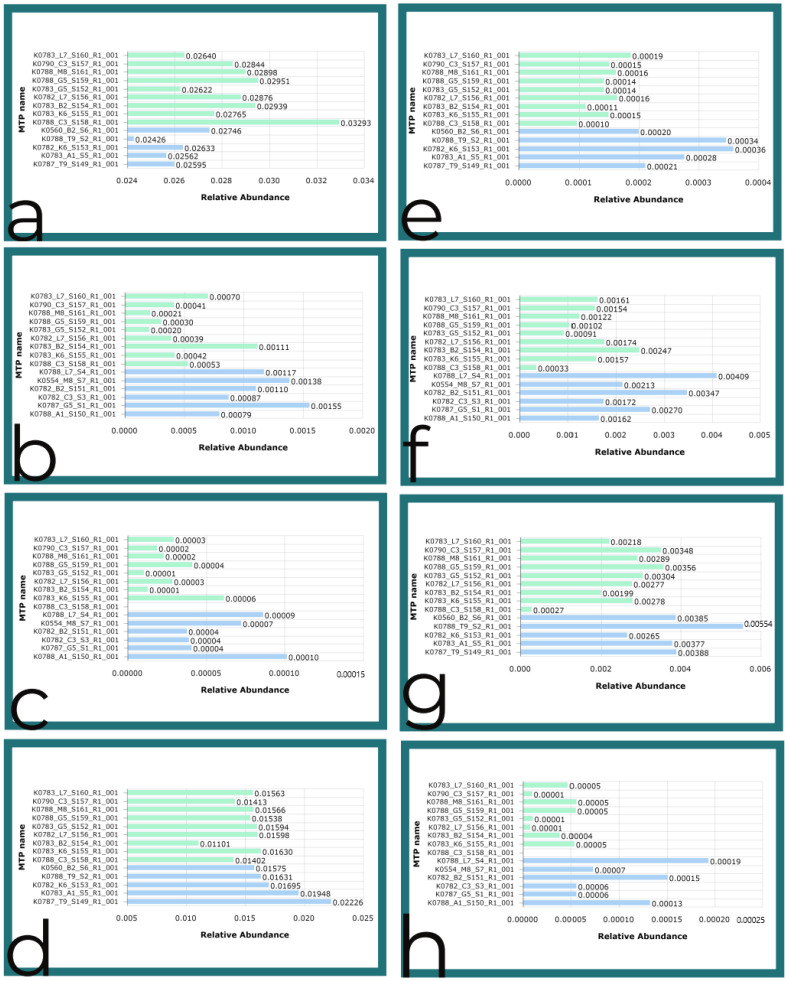
Functional analysis with the Kruskal–Wallis test: (**a**) Assimilatory nitrate reductase for the Green Trail vs. Red Trail; (**b**) Benzene toluene chlorobenzene dioxygenase ferredoxin component for the Green Trail vs. Blue Trail; (**c**) Beta-lactamase for the Green Trail vs. Blue Trail; (**d**) Halo acetate dehalogenase for the Blue Trail vs. Red Trail; (**e**) Propanediol dehydratase for the Green Trail vs. Red Trail; (**f**) Putrescine ornithine dehydratase for the Green Trail vs. Blue Trail; (**g**) Stigmatellin polyketide synthase for the Green Trail vs. Red Trail; (**h**) Lichenysin synthetase for the Green Trail vs. Blue Trail.

**Figure 6 microorganisms-10-01218-f006:**
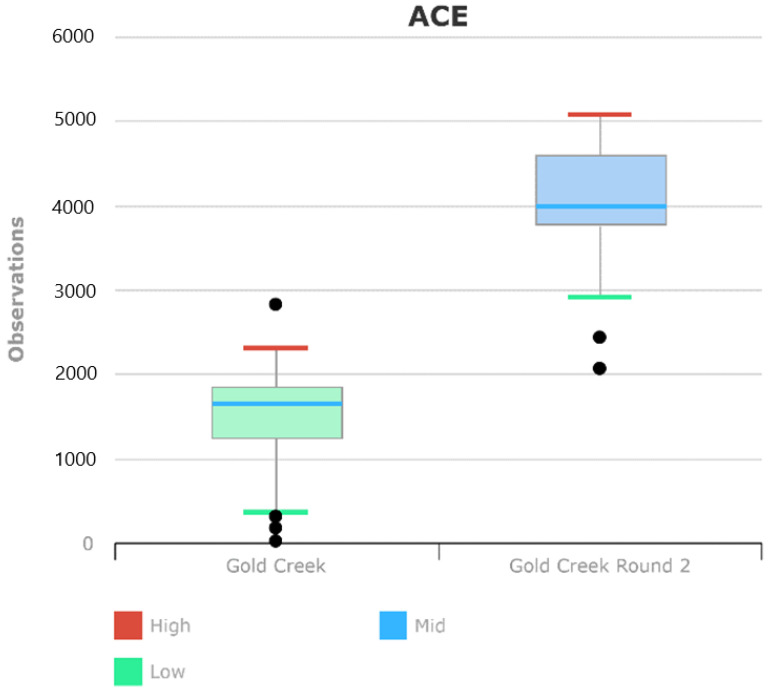
Alpha Diversity Index for the winter versus the summer samples. Note evidence of overdispersion since both groups show outliers.

**Figure 7 microorganisms-10-01218-f007:**
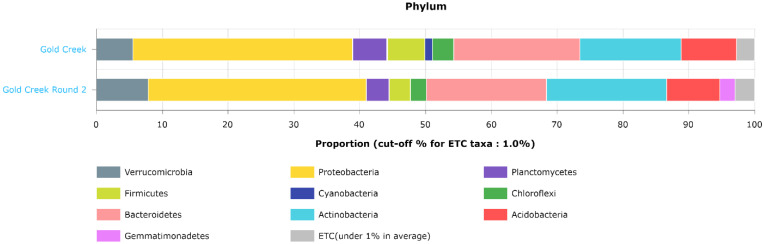
Comparison of the taxonomic composition of the winter sequences versus the summer sequences at the phylum level using EZBioCloud MTP.

**Figure 8 microorganisms-10-01218-f008:**
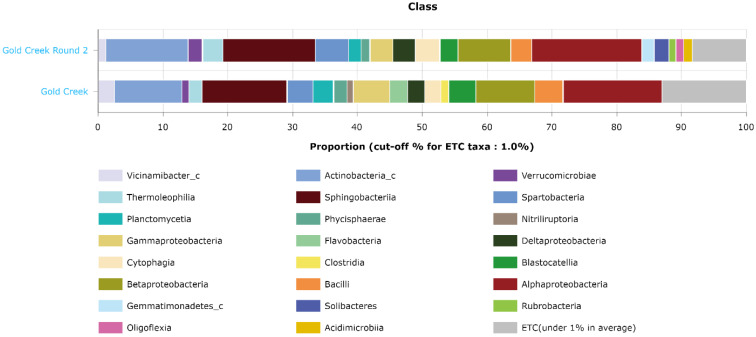
EZBioCloud MTP analysis of the composition of the winter and summer sequences at the class level.

**Figure 9 microorganisms-10-01218-f009:**
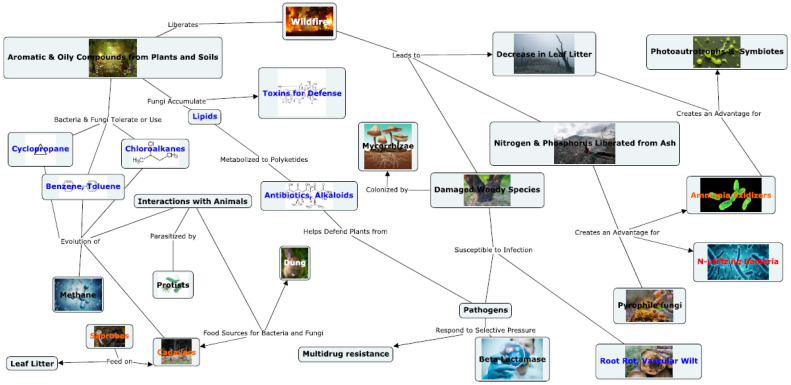
A proposed Soil Food Web of the Gold Creek Preserve, showing the most important redundant functions and known species interactions.

**Table 1 microorganisms-10-01218-t001:** Description of the fires that have affected the area of interest [17]. © 1998–2019 Given Place Media, publishing as Los Angeles Almanac. Reprinted with permission.

Fire Name	Date Started	Acres Burned	Structures Destroyed
Station	26 August 2009	160,577	209
Sand	22 July 2016	41,432	18
Creek	5 December 2017	15,619	123

**Table 2 microorganisms-10-01218-t002:** Soil classifications, soil physical and chemical properties are specified for the Gold Creek Preserve according to USDA historical data [3].

Soil Class	Slope	pH	CEC	Texture	%OM	Avail. H_2_O/100 cm
Trigo-Modesto-San Andreas (Map unit 48)	15–70%	7	9.8	Loam	1	11.27 cm
Caperton-Trigo (Map unit 54)	50–85%	6.5	11.3	Gravelly loam	1	5.94 cm

**Table 3 microorganisms-10-01218-t003:** Sample metadata for Round 2 sampling.

SampleID	Latitude	Longitude	Elevation	Plant_Species	Trail	Replicate
Sample25	34.322778	−118.31167	2140	Charred_Oak	BlueTrail	rep1
Sample26	34.322778	−118.31167	2140	Charred_Oak	BlueTrail	rep2
Sample27	34.322778	−118.31167	2140	Charred_Oak	BlueTrail	rep3
Sample28	34.321389	−118.30889	2240	Charred_Oak	RedTrail	rep1
Sample29	34.321389	−118.30889	2240	Charred_Oak	RedTrail	rep2
Sample30	34.321389	−118.30889	2240	Charred_Oak	RedTrail	rep3
Sample31	34.321667	−118.31111	2200	Manzanita_SW	RedTrail	rep1
Sample32	34.321667	−118.31111	2200	Manzanita_SW	RedTrail	rep2
Sample33	34.321667	−118.31111	2200	Manzanita_SW	RedTrail	rep3
Sample34	34.3267402	−118.30997	2340	Bush_Poppy	GreenTrail	rep1
Sample35	34.3267402	−118.30997	2340	Bush_Poppy	GreenTrail	rep2
Sample36	34.3267402	−118.30997	2340	Bush_Poppy	GreenTrail	rep3
Sample37	34.3279066	−118.31028	2350	Charred_Oak	GreenTrail	rep1
Sample38	34.3279066	−118.31028	2350	Charred_Oak	GreenTrail	rep2
Sample39	34.3279066	−118.31028	2350	Charred_Oak	GreenTrail	rep3
Sample40	34.324444	−118.31167	2190	Yerba_Santa	GreenTrail	rep1
Sample41	34.324444	−118.31167	2190	Yerba_Santa	GreenTrail	rep2
Sample42	34.324444	−118.31167	2190	Yerba_Santa	GreenTrail	rep3

**Table 4 microorganisms-10-01218-t004:** Sample metadata from Round 1 sampling.

Sample Name	Lat	Long	Elevation	Plant	Trail	Type	Topography	Soil Class
K0554_M8	34.3225	−118.313	2140	Oak	Blue Trail	soil	Flat	Trigo-Modesto-San Andreas
K0782_B2	34.32278	−118.313	2130	None	Blue Trail	sediment	Creek	Trigo-Modesto-San Andreas
K0782_C3	34.3225	−118.313	2140	Sycamore	Blue Trail	soil	Flat	Trigo-Modesto-San Andreas
K0787_G5	34.3225	−118.301	2090	Fern	Blue Trail	soil	Flat	Trigo-Modesto-San Andreas
K0788_A1	34.3225	−118.314	2090	Yucca	Blue Trail	soil	Flat	Trigo-Modesto-San Andreas
K0788_L7	34.3225	−118.314	2090	Low Creek	Blue Trail	sediment	Creek	Trigo-Modesto-San Andreas
K0782_L7	34.32389	−118.312	2210	Yerba Santa	Green Trail	soil	Terrace	Trigo-Modesto-San Andreas
K0783_B2	34.3251	−118.54	2230	Reed	Green Trail	sediment	Creek	Caperton-Trigo
K0783_G5	34.32611	−118.31	2330	Chemise	Green Trail	soil	Terrace	Caperton-Trigo
K0783_K6	34.32667	−118.309	2380	Ceanothus	Green Trail	soil	Terrace	Caperton-Trigo
K0783_L7	34.32722	−118.311	2400	Bush Poppy	Green Trail	soil	Terrace	Caperton-Trigo
K0788_C3	34.3251	−118.54	2230	None	Green Trail	sediment	Canyon	Caperton-Trigo
K0788_G5	34.32694	−118.31	2350	None	Green Trail	soil	Canyon	Caperton-Trigo
K0788_M8	34.32389	−118.312	2190	Yerba Santa	Green Trail	soil	Slope	Trigo-Modesto-San Andreas
K0790_C3	34.32694	−118.31	2350	Oak	Green Trail	sediment	Creek	Caperton-Trigo
K0560_B2	34.32194	−118.311	2200	Manzanita	Red Trail	soil	Slope	Trigo-Modesto-San Andreas
K0782_K6	34.32139	−118.309	2240	Oak	Red Trail	soil	Terrace	Trigo-Modesto-San Andreas
K0783_A1	34.3225	−118.312	2160	Toyon	Red Trail	soil	Slope	Trigo-Modesto-San Andreas
K0787_T9	34.32167	−118.311	2210	Manzanita	Red Trail	soil	Slope	Trigo-Modesto-San Andreas
K0788_T9	34.32278	−118.312	2190	Buckwheat	Red Trail	soil	Slope	Trigo-Modesto-San Andreas

**Table 5 microorganisms-10-01218-t005:** DESeq2 results for differentially expressed fungal taxa associated with the Red Trail vs. the Blue trail during winter 2020. Positive log fold change results represent sequences that were differentially abundant on the Red Trail. Negative log fold changes represent sequences that were differentially abundant on the Blue Trail.

	Taxon	BaseMean	log2FoldChange	lfcSE	*p* Value	*p* adj	Notes
OTU667	*Mucor saturninus*	11.66055	−22.2132	4.0127	3.10 × 10^−8^	4.01 × 10^−6^	Produces gamma linoleic acid and stores triacyl glycerides [45]
OTU671	Umbelopsidaceae (unclassified)	10.21483	21.992	4.012815	4.24 × 10^−8^	4.40 × 10^−6^	Converts lignocellulosic sugars to lipids [46]
OTU433	*Cladorrhinum* sp.	41.67976	22.62889	3.923235	8.03 × 10^−9^	1.39 × 10^−6^	Wheat endophyte that controls *Fusarium* [47]
OTU341	*Plectosphaerella* sp.	55.81719	−24.4068	4.012054	1.18 × 10^−9^	3.05 × 10^−7^	Root rot [48,49]
OTU536	*Phallus hadriani*	61.05296	−24.5181	4.01204	9.89 × 10^−10^	3.05 × 10^−7^	Dune stinkhorn [50]

**Table 6 microorganisms-10-01218-t006:** DESeq results for differentially expressed fungal taxa associated with the Green Trail vs. the Blue Trail (Round 1). Positive fold change values correspond to sequences that were differentially abundant on the Green Trail; taxa with significant negative fold change values are associated with the Blue Trail.

	Taxon	BaseMean	log2FoldChange	*p* Value	*p* adj	Notes
OTU341	*Plectosphaerella*	55.81719	−25.5472	1.75 × 10^−13^	4.53 × 10^−11^	Root rot [48,49]
OTU672	*Umbelopsis* sp.	47.74795	−25.354	2.65 × 10^−13^	4.58 × 10^−11^	Endophyte, Converts lignocellulosic sugars to lipids [46]
OTU536	*Phallus hadriani*	61.05296	−25.681	1.31 × 10^−13^	4.53 × 10^−11^	Dune stinkhorn [50]
OTU433	*Cladorrhinum* sp.	41.67976	22.38416	7.61 × 10^−11^	5.57 × 10^−9^	Produces antibiotics [47]
OTU363	*Emericellopsis* sp.	12.82686	−23.4318	1.42 × 10^−11^	1.80 × 10^−9^	Halotolerant, sexual state of Acremonium [51]
OTU224	*Calloria urticae*	10.1225	−22.1403	1.77 × 10^−10^	1.02 × 10^−8^	Pezizalaceae
OTU667	*Mucor saturninus*	11.66055	−23.3332	1.73 × 10^−11^	1.80 × 10^−9^	Produces gamma linoleic acid and stores triacyl glycerides [45]
OTU434	*Cladorrhinum flexuosum*	17.64756	−22.5163	8.61 × 10^−11^	5.57 × 10^−9^	Wheat endophyte that controls *Fusarium* [47]
OTU671	Umbelopsidaceae (unclassified)	10.21483	−22.9432	3.74 × 10^−11^	3.23 × 10^−9^	Converts lignocellulosic sugars to lipids [46]

**Table 7 microorganisms-10-01218-t007:** Red Trail vs. Green Trail 16S Round 2 differential abundance analysis (summer). The negative fold change values represent taxa that are elevated on the Green Trail. The positive fold change values represent taxa that were elevated on the Red Trail.

	Taxon	BaseMean	log2FoldChange	*p* Value	*p* adj	Notes
OTU349	*Gemmatimonas* sp.	10.973	−6.21152	1.50 × 10^−6^	0.000307	Facultative anoxygenic phototroph, requires organic substrates [54]
OTU206	Sphingobacteriaceae (unclassified)	27.15296	−7.84103	1.53 × 10^−8^	9.40 × 10^−6^	Polyketides, Terpenes, non-ribosomal peptides, antibiotics [55]
OTU481	Rickettsiales (unclassified)	62.31641	4.675178	6.40 × 10^−5^	0.009801	Related to Typhus, obligate intracellular parasite of animals [56]
OTU454	Hyphomonadaceae (unclassified)	20.30979	−7.4021	1.44 × 10^−6^	0.000307	Vascular wilt of poplar [57]

**Table 8 microorganisms-10-01218-t008:** Red Trail vs. Blue Trail 16S Round 2 differential abundance analysis. Taxa with negative fold change values were elevated on the Blue Trail. Taxa with positive fold change values were elevated on the Red Trail.

	Taxon	BaseMean	log2FoldChange	*p* Value	*p* adj	Notes
OTU206	Sphingobacteriaceae (unclassified)	27.15296	−6.56403	0.000128	0.003176	Polyketides, Terpenes, antibiotics [55]
OTU349	*Gemmatimonas* sp.	10.973	−6.55623	2.98 × 10^−5^	0.000894	Autotrophic bacterium [54]
OTU481	Rickettsiales (unclassified)	62.31641	7.659409	3.13 × 10^−5^	0.000894	Intracellular parasite [56]
OTU439	*Aminobacter* sp.	30.8104	8.159297	3.77 × 10^−5^	0.000998	Mineralizes Chlorobenzoates [58]
OTU265	Ktedonobacteraceae (unclassfied)	152.1212	8.940275	7.98 × 10^−6^	0.000329	Forms branched mycelia, geothermal areas [59]
OTU512	*Burkholderia* sp.	69.31398	9.189518	1.67 × 10^−6^	7.84 × 10^−5^	Possible plant pathogen and antibiotic resistant, degrades chlorinated pollutants [33]
OTU95	Legionellaceae (unclassified)	121.8497	9.92243	3.03 × 10^−9^	5.63 × 10^−7^	Amino acids are preferred energy source [60]
OTU121	Pseudonocardiaceae (unclassified)	99.14723	10.08414	1.15 × 10^−8^	1.43 × 10^−6^	Degradation of polysaccharides and chitin in dryland soils [61]
OTU432	Methylocystaceae (unclassified)	17.88785	18.59022	1.69 × 10^−6^	7.84 × 10^−5^	Methanotrophic bacteria [62]
OTU89	*Curtobacterium* sp.	17.36935	20.60778	3.87 × 10^−7^	2.87 × 10^−5^	Degradation of polysaccharides, most common in S. CA leaf litter [63]
OTU4	*Thermoplasmata* E2 (unclassified)	20.76215	20.82379	3.36 × 10^−7^	2.87 × 10^−5^	Uncultured archaea, Methanogen [64]
OTU17	Acidobacteriaceae (Unclassified)	48.96786	21.70634	1.43 × 10^−9^	5.31 × 10^−7^	Plant growth-promoting [65]
OTU602	Pseudomonadaceae (unclassified)	45.49218	22.03954	8.96 × 10^−7^	5.54 × 10^−5^	Plant-associated, motile [66]

## Data Availability

Sequence read archives and metadata are available on the National Center for Biotechnology Information, BioProject ID PRJNA831829: http://www.ncbi.nlm.nih.gov/bioproject/831829 (accessed on 29 May 2022).

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
