# Peer review of "The Functional Biogeography of eDNA Metacommunities in the Post-Fire Landscape of the Angeles National Forest"

_microorganisms, 2022, doi:10.3390/microorganisms10061218_

Round 1

Reviewer 1 Report

This manuscript "The Functional Biogeography of eDNA Metacommunities in the Post-fire Landscape of the Angeles National Forest" is interesting piece of work to be considered in microorganisms but it needs some revisions before it is accepted. The comments and suggestions are annotated in the manuscript. 

Author Response

Thank you for your comments, which have improved this manuscript. 

I have appended new versions of Figure 5 and Figure 10 in an effort to improve the resolution of the published images. Scientific names have been italicized appropriately throughout, but not above the genus level. Furthermore, the abbreviation for species (sp.) is no longer italicized.  The keywords are now in alphabetical order. The author list has been updated.

The first line of the introduction was updated to remove the typo.  Reference 8 is now properly cited. The title to Materials and methods section was formatted. The margin formatting for lines 251-256 was updated.

A few updates to the methods section have been provided with additional details. 

Reference #7 was updated with a dash for page ranges.  

Reviewer 2 Report

The research carried out is very interesting and provides a lot of information about the changes in soil microbiome after the fire.

The originality, novelty and quality of the article are appropriate.

However, I would like to make some comment.

For example: Line 27 and 285: ,,pathogenic Fusarium”; how do you know they are pathogenic? You may not know if it is a pathogenic species without measurement its pathogenicity. I think it would be necessary to use the term possibly pathogenic before any pathogens mentioned.

Author Response

Thank you for your comments, which have improved this manuscript. All instances related to pathogens in the results have been amended as possible, potential, or putative pathogens. 

Reviewer 3 Report

After going through the manuscript written by Senn et al. I found it scientifically sound, approprite, and well written; therefore, I would recommend the publication of this manuscript without any specific comments.

Author Response

Thank you for your comments and for taking the time to review our manuscript.